# Design, Synthesis and Bioactive Evaluation of Oxime Derivatives of Dehydrocholic Acid as Anti-Hepatitis B Virus Agents

**DOI:** 10.3390/molecules25153359

**Published:** 2020-07-24

**Authors:** Zhuocai Wei, Jie Tan, Xinhua Cui, Min Zhou, Yunhou Huang, Ning Zang, Zhaoni Chen, Wanxing Wei

**Affiliations:** 1College of Chemistry and Chemical Engineering, Guangxi University, Nanning 530004, China; liveupto123@163.com (Z.W.); yulitanjie@163.com (J.T.); cuixinhua123@163.com (X.C.); leelmm@gxu.edu.cn (M.Z.); 1814302009@st.gxu.edu.cn (Y.H.); 2Nanning Center for Disease Control and Prevention, Nanning 530028, China; 3School of Basic Medical Science, Guangxi Medical University, Nanning 530021, China; zangninggxnn@163.com; 4Pharmaceutical College, Guangxi Medical University, Nanning 530021, China; wonderyao@163.com

**Keywords:** dehydrocholic acid, oxime derivatives, anti-HBV, designation, synthesis

## Abstract

Oxime derivatives of dehydrocholic acid and its esters were designed for anti-hepatitis B virus (HBV) drugs according to principles of assembling active chemical fragments. Twelve compounds were synthesized from dehydrocholic acid by esterification and oxime formation, and their anti-hepatitis B virus (HBV) activities were evaluated with HepG 2.2.15 cells. Results showed that 5 compounds exhibited more effective inhibition of HBeAg than positive control, among them **2b-3** and **2b-1** showed significant anti-HBV activities on inhibiting secretion of HBeAg (IC_50 (**2b-3**)_ = 49.39 ± 12.78 μM, SI _(**2b-3**)_ = 11.03; IC_50 (**2b-1**)_ = 96.64 ± 28.99 μM, SI _(**2b-1**)_ = 10.35) compared to the Entecavir (IC_50_ = 161.24 μM, SI = 3.72). Molecular docking studies showed that most of these compounds interacted with protein residues of heparan sulfate proteoglycan (HSPG) in host hepatocyte and bile acid receptor.

## 1. Introduction

Hepatitis B is a potentially life-threatening liver infection caused by the hepatitis B virus (HBV), and still a major global health problem, which causes chronic infection, and puts people at a high risk of death from cirrhosis and liver cancer [1,2]. According to World Health Organization (WHO) reports, an estimated 257 million people are living with the hepatitis B virus infection in the world (defined as hepatitis B surface antigen positive). In 2015, hepatitis B resulted in 887,000 deaths, mostly from complications (including cirrhosis and hepatocellular carcinoma) [3]. The nucleos(t)ide analogues are recommended for the treatment of chronic hepatitis B in the current consensus guidelines due to their significant suppression of HBV replication [4,5,6,7]. Unfortunately, this treatment is not satisfactory due to the limitations and side effects of nucleos(t)ide drugs. HBV therapy with nucleoside analogs, in long term, has developed resistance and obvious decreased inhibition effects [8,9,10]. The disadvantages of nucleoside analogs prompted us and other researchers to invent and find new structural non-nucleoside analog compounds [11,12,13,14,15,16]. Many anti-HBV bioactive non-nucleoside analog compounds have been designed and developed on the basis of their interactions with receptor using molecular docking [17,18,19,20,21]. When HBV receptor binding domain PreS1 and PreS2 protein (including L protein, M protein and S protein) interact with small molecules, the virus will not allow entry to hepatocyte. A receptor, heparan sulfate proteoglycan (HSPG), which is critical for virus attachment and helps enrich virions on the cell surface (bringing them in close proximity to the receptor) from pre-S2 in hepatocyte interacts with small molecules; the virus will be inhibited to attach the hepatocyte and endocytosis will not be allowed [22]. Cholic acids and their derivatives were reported as liver-targeted vehicles for drug delivery due to their existence in the liver with no side effects and metabolization through enterohepatic circulation [23,24,25,26]. Dehydrocholic acid (DHCA) is a derivative of cholic acid [27] containing one carboxyl group and three carbonyl groups. Drugs targeted and concentrated to the liver organ will increased their anti-HBV effectivity and decreased side effects. Therefore DHCA, with reactive functional groups, was considered as a liver-target vehicle to deliver drugs to the liver in our present work. Based on the bioactivities of DHCA and oximes in our previous works [28,29,30], the introduction of the oxime group to DHCA should be suggested to possess liver-targeted and anti-HBV activities. So, a series of oxime derivatives of DHCA were designed, synthesized, and screened for anti-HBV activity in vitro in this work, and molecular docking studies were carried out to investigate the relationship of structure and bioactivity of these compounds using a molecular operating environment (MOE).

## 2. Results and Discussion

### 2.1. Chemistry

The synthetic layout of the intermediates and target compounds is presented in Scheme 1. In the initial step, ester of DHCA 3,7,12-trioxocholanoate (**1a**, **1b** or **1c**) was prepared by DHCA with alcohol (benzyl alcohol, piperonyl alcohol, or furfuryl alcohol) in the presence of *N*,*N*’-Dicyclohexylcarbodiimide (DCC) in yield of 75.6~76.7%. Then, the intermediates (**1a**, **1b** and **1c**) and DHCA reacted with hydroxylamine, O-benzylhydroxylamine, and methoxyamine hydrochloride in DCM with the presence of sodium acetate trihydrate to afford oxime derivatives of DHCA (**2a-1**, **2a-2**, **2a-3**, **2b-1**, **2b-2**, **2b-3**, **2c-1**, **2c-2**, **2c-3**, **0-1**, **0-2**, and **0-3**) as a mixture of isomers. The isomers were easily separated by column chromatography, and the major isomer of these oxime compounds is presumed to have the *E* configuration [31,32,33]. Structures of these synthesized compounds were elucidated by ^1^H NMR, ^13^C NMR, and MS methods. Signals at 211.91~211.96, 209.06~209.09, and 208.71~208.72 ppm in ^13^C NMR spectra of 3, 7, 12-trioxocholanoate were related to the three carbons of C_12_=O, C_7_=O and C_3_=O groups, respectively, in the structure of steroid [34]. When C_3_=O group was converted to oxime, signals at 208.71~208.72 ppm disappeared and replaced with signals at 156.80~159.40 ppm (C_3_=N) in the ^13^C NMR spectra of these compounds, and a broad singlets belonging to the N-OH, N-OCH_2_, and N-OCH_3_ groups were observed in a range of 7.67~10.23, 4.86~5.00, and 3.68~3.81 (*δ* in ppm), respectively. The signals at 7.43~6.36 ppm were attributed to protons in the furan ring, 7.01~5.95 ppm to protons in piperonyl group, and 7.54~7.26 ppm to protons in benzyl ring.

### 2.2. Anti-Hepatitis B Virus (HBV) Activity

A series of oxime derivatives of DHCA were synthesized, and their anti-HBV activities were evaluated with Entecavir (ETV) as the positive control in vitro on inhibiting the secretion of HBeAg and HBsAg in HepG2.2.15 cells. Results (Table 1) showed that all compounds were more effective for inhibiting secretion of HBeAg than that of HBsAg. Compounds with significant inhibition of HBeAg secretion were compound **2b-3** (IC_50_ = 49.39 ± 12.78 μM, SI = 11.03) and **2b-1** (IC_50_ = 96.64 ± 28.99 μM, SI = 10.35) compared to the reference ETV (IC_50_ = 246.87 ± 50.03 μM, SI = 2.43), and low cytotoxicity against HepG2.2.15 cell lines (CC_50 (**2b-3**)_ = 544.73 ± 28.92, CC_50 (**2b-1**)_ > 1000 μM). The other 3 compounds, **2a-1**, **2c-1**, and **0-2**, showed more effective inhibition of HBeAg than ETV’s. Apart from these 5 compounds mentioned above, other compounds showed ineffective inhibition of HBeAg secretion, not only with low IC_50_ values, but also their high cytotoxicity (SI < 1). Only compounds **2a-1** and **2c** group exhibited weak inhibition of HBsAg secretion, and they did not effectively suppress HBV due to their high toxicity (IS < 1).

### 2.3. Structure–Activity Relationship (SAR)

Results of the bioactive assay showed that DHCA exhibited no activity against the secretion of HBeAg and HBsAg, while most of the derivatives exhibited more or less activity against the secretion of HBsAg and HBeAg, as shown in Table 1.

Five compounds **2a-1**, **2b-1**, **2c-1**, **2b-3**, and **0-2** more effectively inhibited secretion of HBeAg than the positive control ETV. Docking results showed that **2b-1**, **2b-3**, and **0-2** interacted with the HSPG protein with significant docking scores, meanwhile **2a-1** and **2c-1** did not interact with the protein.

Compound **2c-1**, **2c-2**, and **2c-3**, with similar skeletons, but different oxime groups, showed slightly different anti-HBV activity and cytotoxicity. Although with lower IC_50_ values for inhibiting secretion of HBeAg than oxime **2c-1** and benzyl oxime ether **2c-2**, methyl oxime ether **2c-3** was not an effective inhibitor because of its cytotoxicity (SI = 0.86) (Figure 1). All compounds proved ineffective on inhibiting HBsAg secretion (Figure 2). Bioactive screening results revealed that benzyl oxime ethers (**2a-2**, **2b-2**, and **2c-2**) were less effective (on the base of IC_50_ values) in inhibiting HBeAg than oximes and methyl oxime ethers. Introduction of a larger group of benzyl to form oxime ethers decreased their inhibition of HBeAg. The results also revealed that oxime derivatives of DHCA eaters (**2a**, **2b**, and **2c**) more effectively inhibited the secretion of HBeAg than the oxime compounds of DHCA (**0-1**, **0-2**, and **0-3**). Results (Table 1) showed that oximes (R_2_ = OH), **2a-1**, **2b-1**, and **2c-1** were less toxic in groups **2a**, **2b**, and **2c**, respectively. Methyl ether of oximes **2a-3**, **2b-3**, and **2c-3** were more toxic than others in their groups, respectively. Compound group **c** (**2c-1**, **2c-2**, and **2c-3**), O-furan-2-ylmethyl esters, exhibited obvious more cytotoxicity than benzyl ester (group **2a**), piperonyl esters (group **2b**), and DHCA of oximes (group **0**). These results indicated that formation of O-furan-2-ylmethyl esters increased cytotoxicity of oxime ethers.

### 2.4. Molecular Docking Study

To further investigating relationship of structures of the bioactivity and interactions between the ligand and protein of these oxime derivatives, docking studies were carried out using MOE 2008.10. The “Site Finder” tool in this program was used to reach for the active site. Docking study of these oxime derivatives with bile acid receptor protein residue (PDB: 3bej) and HSPG protein residue (PDB: 3sh5) was achieved. The docking scores (S) and the hydrogen bond strength of all the molecules are shown in Appendix A.

Docking studies showed that DHCA and cholic acid interacted to bile acid receptor with scores in −11.35 and −12.35 kcal/mol, respectively, and bound to residues Lys321 and Ile468 (Figure 3). These results confirmed DHCA possessed hepatocyte targeting activity, theoretically. These oxime derivatives had strong interaction with the dock score ranging from −13.46 to −10.85 kcal/mol with bile acid receptor (Appendix A), and −12.54 to −8.43 kcal/mol with HSPG (Appendix A), which was very close to the case of cholic acid and DHCA. Moreover, most of the compounds were involved in at least one hydrogen-bonding interaction with bile acid receptor and HSPG. 

Bioactive results in vitro showed the compounds **2b-3** and **2b-1** had the most potent anti-HBV activity with IC_50_ values of 49.39 ± 12.78 and 96.64 ± 28.99 µM for HBeAg, and corresponding SI values of 11.03 and 10.35, respectively. The docking results of protein residue of the bile acid receptor with compounds **2b-3** and **2b-1** showed formation of compound **2b-3** hydrogen bonds length in 2.68 and 3.36 Å for each oxygen atom in the piperonyl group, with Arg 686 and Tyr 397, and 3.00 and 2.83 Å for the oxygen atom of C_24_=O and C_12_=O in the ether group and carbonyl group, with Ser 392 and Asn 444 (Figure 4). The results also showed formation of compound **2b-1** hydrogen bonds length in 3.54 and 2.85 Å for one oxygen atom of piperonyl group with Tyr 397 and Gln 396, and 2.92 Å for oxygen atom of C_12_=O with His 447 (Figure 5). Results of the docking protein residue of HSPG with compound **2b-3** and **2b-1** showed formation of hydrogen bonds length in 2.68 Å for oxygen atom of C_12_=O of compound **2b-3** with Lys 133, 2.15 Å and 3.16 Å for one oxygen atom of the piperonyl group, with Arg 104, and 2.83 Å for the other oxygen atom of piperonyl group with Val 30 (Figure 6). Similarly, two hydrogen bonds, length 2.31 and 3.25 Å for one oxygen atom of piperonyl group with Arg 104, and 3.22 Å for the other oxygen atom of piperonyl group with Val 30, were found in compound **2b-1** docking results (Figure 7). Contrasting with the **2b-1**, **2b-3**, and **0-2** compounds, **2a-1** and **2c-1** with effective inhibition of HBV showed no interaction to the HSPG protein. These docking results did not coincide with anti-HBV activities. Docking results revealed strong interaction between these oxime derivatives of DHCA and the bile acid receptor, which implied these DHCA derivatives might concentrate to hepatocyte, and possess liver-target activity. Although DHCA and cholic acid docked to the HSPG protein, and showed moderate interactions, they did not exhibit inhibition of HBV in vitro assay (Appendix A.

## 3. Methods

### 3.1. Synthesis Methods

#### 3.1.1. Chemistry and Chemical Methods

Melting points (mp) were determined on a WRX-4 electrothermal melting point apparatus (Shanghai, China) and were uncorrected. The ^1^H NMR and ^13^C NMR spectra were recorded on a Bruker AV III HD 600 MHz spectrometer using CDCl_3_ or DMSO-*d*_6_ as solvent. Chemical shifts were expressed relative to tetramethylsilane (TMS) used as an internal standard and were reported as *δ* (ppm). The mass spectra were taken on Thermo Scientific ITQ 1100 instrument (Thermo Fisher Scientific, Waltham, MA, USA) with an EIS source and an ion trap analyzer in the positive ion mode. All the reactions were monitored by thin-layer chromatography (TLC) on Silica gel GF-254 plates and the products were separated by flash column chromatography on Silica gel H (Qingdao Haiyang Chemical, Qingdao, China).

#### 3.1.2. General Procedure for the Intermediate Compounds (**1a**, **1b**, and **1c**)

DHCA (1 equiv., 1.25 mmol) and benzyl alcohol, 1,3-benzodioxole-5-methanol, or furfuryl alcohol (1 equiv., 1.25 mmol) were added to a solution of the 4-dimethylaminopyridine (DMAP, 5%) in CH_2_Cl_2_ (15 mL). The mixture was stirred and cooled to 0 °C for 0.5 h and then *N, N*-dicyclohexylcarbodiimide (DCC) (1 equiv., 1.25 mmol) was added over a 5-min period. Finally, the reaction was stirred under anhydrous conditions for overnight at room temperature. After stopping, deionized water (4 mL) was added to the resultant solution with lots of white solid (1,3-dicyclohexylurea, DCU) appearing. The mixture was filtered and the filtrate was evaporated to yield a crude product which was purified by column chromatography eluting with an eluent of ethyl acetate/petroleum ether (1:2, *v/v*) to afford **1a**, **1b** and **1c**, respectively.

*Benzyl 3,7,12-trioxocholanoate* (**1a**). White crystal, yield 76.5%, mp 203.8~205.8 °C. ^1^H NMR (600 MHz, CDCl_3_) *δ* 7.44–7.28 (5H, m, H-27, 28, 29, 30, 31), 5.13 (2H, q, *J* = 12.3 Hz, H-25), 2.99–2.81 (3H, m, H-6α, 8β, 23), 2.53–1.81 (16H, m, H-1α, 2β, 4, 5, 6β, 9, 11, 14, 15α, 16β, 17, 22, 23), 1.62 (1H, td, *J* = 14.5, 4.5 Hz, H-20), 1.48–1.22 (7H, m, H-1β, 2α, 15β, 16α, 18), 1.06 (3H, s, H-19), 0.86 (3H, d, *J* = 6.7 Hz, H-21). ^13^C NMR (151 MHz, CDCl_3_) *δ* 211.96 (C-12), 209.09 (C-7), 208.72 (C-3), 173.88 (C-24), 136.10 (C-26), 128.55 (C-28, 30), 128.27 (C-27, 31), 128.20 (C-29), 66.12 (C-25), 56.90 (C-17), 51.76 (C-14), 49.00 (C-8), 46.86 (C-9), 45.66 (C-13), 45.56 (C-6), 44.99 (C-5), 42.80 (C-4), 38.63 (C-1), 36.49 (C-2), 36.02 (C-11), 35.46 (C-10), 35.29 (C-20), 31.56 (C-23), 30.46 (C-22), 27.60 (C-15), 25.14 (C-16), 21.92 (C-19), 18.62 (C-21), 11.83 (C-18). ESIMS: *m/z* 515.2774 [M + Na]^+^, calc. for C_31_H_40_O_5_ (492.2876).

*3,4-Methylenedioxybenzyl 3,7,12-trioxocholanoate* (**1b**). White crystal, yield 75.6%, mp 209.6~210.6 °C. ^1^H NMR (600 MHz, CDCl_3_) *δ* 6.95–6.72 (3H, m, H-27, 28, 31), 5.98 (2H, s, H-32), 5.02 (2H, q, *J* = 12.0 Hz, H-25), 2.98–2.81 (3H, m, H-6α, 8β, 23), 2.47–1.82 (16H, m, H-1α, 2β, 4, 5, 6β, 9, 11, 14, 15α, 16β, 17, 22, 23), 1.62 (1H, td, *J* = 14.5, 4.6 Hz, H-20), 1.46–1.21 (7H, m, H-1β, 2α, 15β, 16α, 18), 1.06 (3H, s, H-19), 0.85 (3H, d, *J* = 6.7 Hz, H-21). ^13^C NMR (151 MHz, CDCl_3_) *δ* 211.93 (C-12), 209.07 (C-7), 208.71 (C-3), 173.86 (C-24), 147.79 (C-30), 147.59 (C-29), 129.88 (C-26), 122.28 (C-27), 109.07 (C-31), 108.23 (C-28), 101.16 (C-32), 66.09 (C-25), 56.89 (C-17), 51.75 (C-14), 49.00 (C-8), 46.85 (C-9), 45.66 (C-13), 45.55 (C-6), 44.99 (C-5), 42.80 (C-4), 38.63 (C-1), 36.49 (C-2), 36.02 (C-11), 35.46 (C-10), 35.29 (C-20), 31.56 (C-23), 30.44 (C-22), 27.61 (C-15), 25.13 (C-16), 21.92 (C-19), 18.62 (C-21), 11.81 (C-18). ESIMS: *m/z* 559.2661 [M + Na]^+^, calc. for C_32_H_40_O_7_ (536.2774).

*Furan-2-ylmethyl 3,7,12-trioxocholanoate* (**1c**). White crystal, yield 76.7%, mp 196.5~198.5 °C. ^1^H NMR (600 MHz, CDCl_3_) *δ* 7.40 (1H, d, *J* = 1.7 Hz, H-29), 6.38 (1H, d, *J* = 2.8 Hz, H-27), 6.34 (1H, dd, *J* = 3.2, 1.8 Hz, H-28), 5.04 (2H, q, *J* = 12.3 Hz, H-25), 2.96–2.78 (3H, m, H-6α, 8β, 23), 2.46–1.78 (16H, m, H-1α, 2β, 4, 5, 6β, 9, 11, 14, 15α, 16β, 17, 22, 23), 1.60 (1H, td, *J* = 14.5, 4.5 Hz, H-20), 1.43–1.24 (7H, m, H-1β, 2α, 15β, 16α, 18), 1.03 (3H, s, H-19), 0.81 (3H, d, *J* = 6.7 Hz, H-21). ^13^C NMR (151 MHz, CDCl_3_) *δ* 211.91 (C-12), 209.06 (C-7), 208.71 (C-3), 173.62 (C-24), 149.64 (C-26), 143.20 (C-29), 110.55 (C-28), 110.53 (C-27), 57.87 (C-25), 56.89 (C-17), 51.75 (C-14), 48.99 (C-8), 46.84 (C-9), 45.65 (C-13), 45.54 (C-6), 44.98 (C-5), 42.79 (C-4), 38.63 (C-1), 36.48 (C-2), 36.01 (C-11), 35.45 (C-10), 35.28 (C-20), 31.42 (C-23), 30.38 (C-22), 27.59 (C-15), 25.15 (C-16), 21.91 (C-19), 18.60 (C-21), 11.82 (C-18). ESIMS: *m/z* 505.2559 [M + Na]^+^, calc. for C_29_H_38_O_6_ (482.2668).

#### 3.1.3. General Procedure for the Target Compounds (**2a-1**~**2a-3**, **2b-1**~**2b-3**, **2c-1**~**2c-3**, and **0-1**~**0-3**)

The above intermediate compound **1a**, **1b**, **1c** or DHCA (1 equiv., 0.7mmol) and hydroxylamine, O-Benzylhydroxylamine, or methoxyamine hydrochloride (1.3 equiv., 0.91 mmol) was refluxed for 3~12 h with sodium acetate trihydrate (1 equiv., 0.7 mmol) in CH_2_Cl_2_ (10 mL). After stopping, the mixture was concentrated to give a residue under vacuum. This residue was poured into deionized water with 50% ethanol (30 mL) and extracted with ethyl acetate (2 × 50 mL), and then purified by column chromatography on silica gel eluting with ethyl acetate/petroleum ether (1:6, *v/v*) to give the target compounds.

*Benzyl (E)-3-(hydroxyimino)-7,12-dioxocholanoate* (**2a-1**). White crystal, yield 60.3%, mp 199.2~201.2 °C. ^1^H NMR (600 MHz, CDCl_3_) *δ* 7.85 (1H, s, -C=N-OH), 7.54–7.30 (5H, m, H-27, 28, 29, 30, 31), 5.13 (2H, q, *J* = 12.3 Hz, H-25), 3.15 (1H, d, *J* = 15.0 Hz, H-2β), 2.96–2.76 (3H, m, H-6α, 8β, 23), 2.51–1.77 (14H, m, H-1α, 4, 5, 6β, 9, 11, 14, 16β, 17, 22, 23), 1.69 (1H, td, *J* = 14.9, 4.5 Hz, H-20), 1.40–1.20 (8H, m, H-1β, 2α, 15, 16α, 18), 1.03 (3H, s, H-19), 0.85 (3H, d, *J* = 6.6 Hz, H-21). ^13^C NMR (151 MHz, CDCl_3_) *δ* 212.77 (C-12), 209.18 (C-7), 173.91 (C-24), 158.55 (C-3), 136.11 (C-26), 128.55 (C-28, 30), 128.26 (C-27, 31), 128.19 (C-29), 66.12 (C-25), 56.95 (C-17), 51.95 (C-14), 48.97 (C-8), 46.42 (C-13), 45.60 (C-9), 45.19 (C-6), 44.97 (C-10), 38.48 (C-11), 36.55 (C-20), 35.46 (C-23), 34.53 (C-22), 33.07 (C-4), 31.57 (C-2), 30.47 (C-15), 27.62 (C-1), 25.24 (C-16), 22.24 (C-21), 19.07 (C-5), 18.61 (C-19), 11.79 (C-18). ESIMS: *m/z* 508.3062 [M + H]^+^, 530.2870 [M + Na]^+^, calc. for C_31_H_41_NO_5_ (507.2985).

*3,4-Methylenedioxybenzyl (E)-3-(hydroxyimino)-7,12-dioxocholanoate* (**2b-1**). White crystal, yield 62.1%, mp 182.5~184.5 °C. ^1^H NMR (600 MHz, CDCl_3_) *δ* 7.67 (1H, s, -C=N-OH), 6.99–6.67 (3H, m, H-27, 28, 31), 5.98 (2H, s, H-32), 5.02 (2H, q, *J* = 12.0 Hz, H-25), 3.15 (1H, d, *J* = 15.1 Hz, H-2β), 2.96–2.72 (3H, m, H-6α, 8β, 23), 2.50–1.70 (15H, m, H-1α, 4, 5, 6β, 9, 11, 14, 16β, 17, 20, 22, 23), 1.40–1.23 (8H, m, H-1β, 2α, 15, 16α, 18), 1.04 (3H, s, H-19), 0.85 (3H, d, *J* = 6.7 Hz, H-21). ^13^C NMR (151 MHz, CDCl_3_) *δ* 212.73 (C-12), 209.20 (C-7), 173.92 (C-24), 158.60 (C-3), 147.78 (C-30), 147.58 (C-29), 129.88 (C-26), 122.28 (C-27), 109.07 (C-31), 108.23 (C-28), 101.15 (C-32), 66.09 (C-25), 56.94 (C-17), 51.93 (C-14), 48.97 (C-8), 46.44 (C-13), 45.60 (C-9), 45.20 (C-6), 44.97 (C-10), 38.48 (C-11), 36.55 (C-20), 35.46 (C-23), 34.53 (C-22), 33.06 (C-4), 31.58 (C-2), 30.44 (C-15), 27.62 (C-1), 25.23 (C-16), 22.24 (C-21), 19.06 (C-5), 18.61 (C-19), 11.78 (C-18). ESIMS: *m/z* 552.2943 [M + H]^+^, 574.2767 [M + Na]^+^, calc. for C_32_H_42_NO_7_ (551.2883).

*Furan-2-ylmethyl (E)-3-(hydroxyimino)-7,12-dioxocholanoate* (**2c-1**). White crystal, yield 60.1%, mp 193.2~195.2 °C. ^1^H NMR (600 MHz, CDCl_3_) *δ* 7.72 (1H, s, -C=N-OH), 7.41 (1H, d, *J* = 1.1 Hz, H-29), 6.40 (1H, d, *J* = 3.2 Hz, H-27), 6.36 (1H, dd, *J* = 3.2, 1.9 Hz, H-28), 5.06 (2H, q, *J* = 12.3 Hz, H-25), 3.15 (1H, d, *J* = 15.1 Hz, H-2β), 2.96–2.68 (3H, m, H-6α, 8β, 23), 2.42–1.53(15H, m, H-1α, 4, 5, 6β, 9, 11, 14, 16β, 17, 20, 22, 23), 1.38–1.19 (8H, m, H-1β, 2α, 15, 16α, 18), 1.03 (3H, s, H-19), 0.82 (3H, d, *J* = 6.7 Hz, H-21). ^13^C NMR (151 MHz, CDCl_3_) *δ* 212.38 (C-12), 209.11 (C-7), 173.67 (C-24), 158.19 (C-3), 149.63 (C-26), 143.19 (C-29), 110.54 (C-28), 110.52 (C-27), 57.87 (C-25), 56.87 (C-17), 51.79 (C-14), 48.98 (C-8), 45.61 (C-9), 45.36 (C-13), 45.32 (C-6), 45.05 (C-10), 38.50 (C-11), 36.53 (C-20), 35.63 (C-5), 35.46 (C-23), 31.43 (C-2), 30.39 (C-15), 27.60 (C-1), 26.25 (C-22), 25.64 (C-4), 25.20 (C-16), 22.33 (C-21), 18.60 (C-19), 11.80 (C-18). ESIMS: *m/z* 498.2839 [M + H]^+^, 520.2668 [M + Na]^+^, calc. for C_29_H_39_NO_6_ (497.2777).

*Benzyl (E)-3-((benzyloxy)imino)-7,12-dioxocholanoate* (**2a-2**). White solid, yield 50.9%, mp 135.3~137.3 °C. ^1^H NMR (600 MHz, CDCl_3_) *δ* 7.39–7.26 (10H, m, H-27, 28, 29, 30, 31, 34, 35, 36, 37, 38), 5.11 (2H, q, *J* = 12.3 Hz, H-25), 5.06–5.00 (2H, m, H-32), 3.13 (1H, d, *J* = 14.9 Hz, H-2β), 2.94–2.67 (3H, m, H-6α, 8β, 23), 2.47–1.62 (15H, m, H-1α, 4, 5, 6β, 9, 11, 14, 16β, 17, 20, 22, 23), 1.36–1.16 (8H, m, H-1β, 2α, 15, 16α, 18), 1.01 (3H, s, H-19), 0.83 (3H, d, *J* = 6.7 Hz, H-21). ^13^C NMR (151 MHz, CDCl_3_) *δ* 212.31 (C-12), 209.10 (C-7), 173.88 (C-24), 158.30 (C-3), 138.01 (C-33), 136.12 (C-26), 128.55 (C-35, 37), 128.33 (C-28, 30), 128.27 (C-34, 38), 128.19 (C-29), 127.93 (C-27, 31), 127.69 (C-36), 75.36 (C-32), 66.11 (C-25), 56.88 (C-17), 51.85 (C-14), 48.93 (C-8), 46.57 (C-13), 45.62 (C-9), 45.20 (C-6), 44.99 (C-10), 38.49 (C-11), 36.52 (C-20), 35.47 (C-23), 34.67 (C-22), 33.16 (C-4), 31.57 (C-2), 30.48 (C-15), 27.63 (C-1), 25.22 (C-16), 22.24 (C-21), 20.20 (C-5), 18.62 (C-19), 11.80 (C-18). ESIMS: *m/z* 598.3533 [M + H]^+^, calc. for C_38_H_47_NO_5_ (597.3454).

*3,4-Methylenedioxybenzyl (E)-3-((benzyloxy)imino)-7,12-dioxocholanoate* (**2b-2**). White solid, yield 51.1%, mp 124.3~126.3 °C. ^1^H NMR (600 MHz, CDCl_3_) *δ* 7.39–7.28 (5H, m, H-34, 35, 36, 37, 38, 39), 7.01–6.71 (3H, m, H-27, 28, 31), 5.98 (2H, s, H-32), 5.18–4.86 (4H, m, H-25, 33), 3.15 (1H, d, *J* = 15.0 Hz, H-2β), 2.95–2.74 (3H, m, H-6α, 8β, 23), 2.49–1.64 (15H, m, H-1α, 4, 5, 6β, 9, 11, 14, 16β, 17, 20, 22, 23), 1.39–1.24 (8H, m, H-1β, 2α, 15, 16α, 18), 1.04 (3H, s, H-19), 0.85 (3H, d, *J* = 6.7 Hz, H-21). ^13^C NMR (151 MHz, CDCl_3_) *δ* 212.29 (C-12), 209.10 (C-7), 173.88 (C-24), 158.30 (C-3), 147.79 (C-30), 147.59 (C-29), 138.01 (C-34), 129.89 (C-37), 128.33 (C-36, 38), 127.93 (C-35, 39), 127.69 (C-26), 122.28 (C-27), 109.07 (C-31), 108.23 (C-28), 101.16 (C-32), 75.36 (C-33), 66.08 (C-25), 56.87 (C-17), 51.85 (C-14), 48.93 (C-8), 46.57 (C-13), 45.61 (C-9), 45.20 (C-6), 44.99 (C-10), 38.49 (C-11), 36.52 (C-20), 35.48 (C-23), 34.67 (C-22), 33.16 (C-4), 31.58 (C-2), 30.46 (C-15), 27.63 (C-1), 25.22 (C-16), 22.24 (C-21), 20.20 (C-5), 18.62 (C-19), 11.79 (C-18). ESIMS: *m/z* 664.3219 [M + Na]^+^, calc. for C_39_H_47_NO_7_ (641.3353).

*Furan-2-ylmethyl (E)-3-((benzyloxy)imino)-7,12-dioxocholanoate* (**2c-2**). White solid, yield 51.5%, mp 131.5~133.5 °C. ^1^H NMR (600 MHz, CDCl_3_) *δ* 7.43 (1H, d, *J* = 1.0 Hz, H-29), 7.38–7.28 (5H, m, H-32, 33, 34, 35, 36), 6.42 (1H, d, *J* = 3.1 Hz, H-27), 6.38 (1H, dd, *J* = 2.9, 1.9 Hz, H-28), 5.31–4.80 (4H, m, H-25, 30), 3.15 (1H, d, *J* = 15.1 Hz, H-2β), 2.95–2.72 (3H, m, H-6α, 8β, 23), 2.49–1.64 (15H, m, H-1α, 4, 5, 6β, 9, 11, 14, 16β, 17, 20, 22, 23), 1.40–1.24 (8H, m, H-1β, 2α, 15, 16α, 18), 1.04 (3H, s, H-19), 0.84 (3H, d, *J* = 6.6 Hz, H-21). ^13^C NMR (151 MHz, CDCl_3_) *δ* 212.31 (C-12), 209.12 (C-7), 173.66 (C-24), 158.31 (C-3), 149.65 (C-26), 143.20 (C-29), 138.00 (C-31), 128.33 (C-33, 35), 127.93 (C-32, 36), 127.69 (C-34), 110.55 (C-28), 110.53 (C-27), 75.36 (C-30), 57.87 (C-25), 56.88 (C-17), 51.85 (C-14), 48.93 (C-8), 46.57 (C-13), 45.61 (C-9), 45.19 (C-6), 44.99 (C-10), 38.49 (C-11), 36.52 (C-20), 35.47 (C-23), 34.67 (C-22), 33.15 (C-4), 31.44 (C-2), 30.40 (C-15), 27.61 (C-1), 25.24 (C-16), 22.24 (C-21), 20.20 (C-5), 18.60 (C-19), 11.80 (C-18). ESIMS: *m/z* 588.3301 [M + H]^+^, 610.3130 [M + Na]^+^, calc. for C_36_H_45_NO_6_ (587.3247).

*Benzyl (E)-3-(methoxyimino)-7,12-dioxolcholanoate* (**2a-3**). White solid, yield 61.5%, mp 132.6~134.6 °C. ^1^H NMR (600 MHz, CDCl_3_) *δ* 7.45–7.29 (5H, m, H-27, 28, 29, 30, 31), 5.13 (2H, q, *J* = 12.3 Hz, H-25), 3.81 (3H, s, H-32), 3.06 (1H, d, *J* = 15.1 Hz, H-2β), 2.96–2.72 (3H, m, H-6α, 8β, 23), 2.53–1.64 (15H, m, H-1α, 4, 5, 6β, 9, 11, 14, 16β, 17, 20, 22, 23), 1.40–1.22 (8H, m, H-1β, 2α, 15, 16α, 18), 1.03 (3H, s, H-19), 0.86 (3H, d, *J* = 6.7 Hz, H-21). ^13^C NMR (151 MHz, CDCl_3_) *δ* 212.31 (C-12), 209.12 (C-7), 173.88 (C-24), 157.64 (C-3), 136.12 (C-26), 128.55 (C-28, 30), 128.22 (C-27, 31), 128.19 (C-29), 66.10 (C-25), 61.18 (C-32), 56.88 (C-17), 51.84 (C-24), 48.94 (C-8), 46.64 (C-13), 45.62 (C-9), 45.21 (C-6), 45.00 (C-10), 38.49 (C-11), 36.54 (C-20), 35.47 (C-23), 34.69 (C-22), 33.17 (C-4), 31.57 (C-2), 30.48 (C-15), 27.62 (C-1), 25.21 (C-16), 22.26 (C-21), 19.89 (C-5), 18.62 (C-19), 11.80 (C-18). ESIMS: *m/z* 522.3220 [M + H]^+^, 544.3052 [M + Na]^+^, calc. for C_32_H_43_NO_5_ (521.3141).

*3,4-Methylenedioxybenzyl (E)-3-(methoxyimino)-7,12-dioxocholanoate* (**2b-3**). White solid, yield 62.2%, mp 118.7~120.7 °C. ^1^H NMR (600 MHz, CDCl_3_) *δ* 6.89–6.69 (3H, m, H-27, 28, 31), 5.95 (2H, s, H-32), 5.00 (2H, q, *J* = 12.0 Hz, H-25), 3.78 (3H, s, H-33), 3.03 (1H, d, *J* = 14.9 Hz, H-2β), 2.93–2.70 (3H, m, H-6α, 8β, 23), 2.44–1.62 (15H, m, H-1α, 4, 5, 6β, 9, 11, 14, 16β, 17, 20, 22, 23), 1.39–1.22 (8H, m, H-1β, 2α, 15, 16α, 18), 1.01 (3H, s, H-19), 0.82 (3H, d, *J* = 6.6 Hz, H-21). ^13^C NMR (151 MHz, CDCl_3_) *δ* 212.30 (C-12), 209.12 (C-7), 173.88 (C-24), 157.64 (C-3), 147.78 (C-30), 147.58 (C-29), 129.89 (C-26), 122.27 (C-27), 109.07 (C-31), 108.23 (C-28), 101.15 (C-32), 66.07 (C-25), 61.17 (C-33), 56.88 (C-17), 51.84 (C-14), 48.94 (C-8), 46.63 (C-13), 45.61 (C-9), 45.20 (C-6), 45.00 (C-10), 38.49 (C-11), 36.54 (C-20), 35.47 (C-23), 34.69 (C-22), 33.17 (C-4), 31.57 (C-2), 30.46 (C-15), 27.63 (C-1), 25.21 (C-16), 22.26 (C-21), 19.89 (C-5), 18.62 (C-19), 11.79 (C-18). ESIMS: *m/z* 566.3123 [M + H]^+^, 588.2953 [M + Na]^+^, calc. for C_33_H_43_NO_7_ (565.3040).

*Furan-2-ylmethyl (E)-3-(methoxyimino)-7,12-dioxocholanoate* (**2c-3**). White solid, yield 60.8%, mp 122.3~124.3 °C. ^1^H NMR (600 MHz, CDCl_3_) *δ* 7.43 (1H, d, *J* = 1.2 Hz, H-29), 6.41 (1H, d, *J* = 3.2 Hz, H-27), 6.37 (1H, dd, *J* = 3.1, 1.9 Hz, H-28), 5.07 (2H, q, *J* = 12.3 Hz, H-25), 3.80 (3H, s, H-30), 3.06 (1H, d, *J* = 15.5 Hz, H-2β), 2.96–2.73 (3H, m, H-6α, 8β, 23), 2.48–1.65 (15H, m, H-1α, 4, 5, 6β, 9, 11, 14, 16β, 17, 20, 22, 23), 1.42–1.26 (8H, m, H-1β, 2α, 15, 16α, 18), 1.04 (3H, s, H-19), 0.84 (3H, d, *J* = 6.7 Hz, H-21). ^13^C NMR (151 MHz, CDCl_3_) *δ* 212.31 (C-12), 209.13 (C-7), 173.66 (C-24), 157.65 (C-3), 149.65 (C-26), 143.20 (C-29), 110.55 (C-28), 110.52 (C-27), 61.16 (C-30), 57.87 (C-25), 56.88 (C-17), 51.77 (C-14), 48.94 (C-8), 46.64 (C-13), 45.62 (C-9), 45.20 (C-6), 45.00 (C-10), 38.50 (C-11), 36.54 (C-6), 35.47 (C-23), 34.69 (C-22), 33.17 (C-4), 31.44 (C-2), 30.40 (C-15), 27.61 (C-1), 25.23 (C-16), 22.26 (C-21), 19.89 (C-5), 18.61 (C-19), 11.80 (C-18). ESIMS: *m/z* 534.2811 [M + Na]^+^, calc. for C_30_H_41_NO_6_ (511.2934).

*(E)-3-(hydroxyimino)-7,12-dioxolcholanic acid* (**0-1**). White solid, yield 63.8%, mp 250.4~252.4 °C. ^1^H NMR (600 MHz, DMSO-*d*_6_) *δ* 10.23 (1H, s, -C=N-OH), 3.07–2.78 (4H, m, H-2β, 6α, 8β, 23), 2.35–1.40 (15H, m, H-1α, 4, 5, 6β, 9, 11, 14, 16β, 17, 20, 22, 23), 1.30–1.20 (8H, m, H-1β, 2α, 15, 16α, 18), 0.99 (3H, s, H-19), 0.76 (3H, d, *J* = 5.5 Hz, H-21). ^13^C NMR (151 MHz, DMSO-*d*_6_) *δ* 212.60 (C-12), 210.21 (C-7), 175.31 (C-24), 156.80 (C-3), 56.68 (C-17), 51.78 (C-14), 48.42 (C-8), 46.53 (C-13), 45.82 (C-9), 45.27 (C-6), 45.23 (C-10) 44.63 (C-5), 38.72 (C-11), 36.69 (C-20), 35.46 (C-23), 34.64 (C-22), 33.30 (C-4), 31.59 (C-2), 30.85 (C-15), 27.69 (C-1), 25.07 (C-16), 22.29 (C-21), 19.14 (C-19), 11.89 (C-18). ESIMS: *m/z* 418.2581 [M + H]^+^, 440.2403 [M + Na]^+^, calc. for C_24_H_35_NO_5_ (417.2515).

*(E)-3-((benzyloxy)imino)-7,12-dioxolcholanic acid* (**0-2**). White solid, yield 51.2%, mp 198.5~200.5 °C. ^1^H NMR (600 MHz, DMSO-*d*_6_) *δ* 11.95 (1H, s, -COOH), 7.56–7.00 (5H, m, H-27, 28, 29, 30, 31), 4.97 (2H, s, H-25), 3.06–2.73 (4H, m, H-2β, 6α, 8β, 23), 2.42–1.52 (15H, m, H-1α, 4, 5, 6β, 9, 11, 14, 16β, 17, 20, 22, 23), 1.28–1.15 (8H, m, H-1β, 2α, 15, 16α, 18), 0.99 (3H, s, H-19), 0.76 (3H, d, *J* = 6.1 Hz, H-21). ^13^C NMR (151 MHz, DMSO-*d*_6_) *δ* 212.48 (C-12), 210.12 (C-7), 175.26 (C-24), 159.40 (C-3), 138.77 (C-26), 128.70 (C-28, 30), 128.18 (C-27, 31), 128.05 (C-29), 74.71 (C-25), 56.68 (C-17), 51.70 (C-14), 48.37 (C-8), 46.47 (C-13), 45.83 (C-9), 45.20 (C-6), 44.96 (C-10), 44.29 (C-5), 38.65 (C-11), 36.62 (C-20), 35.47 (C-23), 34.59 (C-22), 33.02 (C-4), 31.56 (C-2), 30.84 (C-15), 27.70 (C-1), 25.11 (C-16), 22.05 (C-21), 19.15 (C-19), 11.88 (C-18). ESIMS: *m/z* 508.3042 [M + H]^+^, 530.2865 [M + Na]^+^, calc. for C_31_H_41_NO_5_ (507.2985).

*(E)-3-(methoxyimino)-7,12-dioxolcholanic acid* (**0-3**). White solid, yield 61.3%, mp 197.2~199.2 °C. ^1^H NMR (600 MHz, DMSO-*d*_6_) *δ* 11.96 (1H, s, -COOH), 3.68 (3H, s, H-25), 3.05–2.73 (4H, m, H-2β, 6α, 8β, 23), 2.41–1.45 (15H, m, H-1α, 4, 5, 6β, 9, 11, 14, 16β, 17, 20, 22, 23), 1.29–1.13 (8H, m, H-1β, 2α, 15, 16α, 18), 0.99 (3H, s, H-19), 0.76 (3H, d, *J* = 5.4 Hz, H-21). ^13^C NMR (151 MHz, DMSO-*d*_6_) *δ* 212.48 (C-12), 210.13 (C-7), 175.30 (C-24), 158.62 (C-3), 60.98 (C-25), 56.68 (C-17), 51.71 (C-14), 48.37 (C-8), 46.51 (C-13), 45.83 (C-9), 45.23 (C-6), 44.99 (C-10), 44.33 (C-5), 38.66 (C-11), 36.65 (C-20), 35.47 (C-23), 34.60 (C-22), 32.98 (C-4), 31.60 (C-2), 30.86 (C-15), 27.70 (C-1), 25.09 (C-16), 22.07 (C-21), 19.15 (C-19), 11.88 (C-18). ESIMS: *m/z* 454.2575 [M + Na]^+^, calc. for C_25_H_37_NO_5_ (431.2672).

### 3.2. Biological Evaluation Methods 

#### 3.2.1. Cell Culture and Drug Treatment

HepG2.2.15 cells were kindly provided by the Chinese Academy of Medical Sciences (Beijing, China), derived from human hepatoma cell line G2. These cells were cultured in Dulbecco’s Modified Eagle’s Medium (DMEM), supplemented with 10% fetal bovine serum (FBS), 1.5 g/L of sodium bicarbonate, 10 mL/L of penicillin, and streptomycin, respectively, and 200 mg/L of G418 at 37 °C under 5% carbon dioxide in a 95–98% humidity. The compounds were diluted to the desired concentrations in culture medium. Before treated with various concentrations of compounds, cells were seeded at a density of 1 × 10^5^ cells/mL in 96-well plates and incubated for 24 h at 37 °C. Every three days in a nine day period, supernatant of each well was replaced with compound-containing fresh medium [23]. 

#### 3.2.2. Method for Cell Toxicity and HBsAg and HBeAg Inhibition Assays 

The levels of HBeAg and HBsAg in the supernatant of the cells were measured using ELISA assay following the manufacturer’s protocol (Shanghai Kehua Bio-engineering Co., Ltd., Shanghai, China). The synthesized compounds were expressed as the concentration that achieved 50% inhibition (IC_50_) to the secretion of HBeAg and HBsAg [29].

The cytotoxicity activity of the synthesized compounds was determined by MTT assay [35]. After refreshing the supernatant, 20 μL MTT (5 mg/mL) was added to each well, which was further cultured for 4 h at 37 °C. Then the supernatant of each well was carefully removed, formazan crystals were dissolved in 150 mL of DMSO and the absorbance at 450 nm was recorded [36]. Cytotoxicity of these compounds was expressed as the concentration of compound required to kill 50% (CC_50_) of the HepG2.2.15 cells.

### 3.3. Molecular Docking

The crystal structures of heparan sulfate proteoglycan (HSPG) protein (P98160, PDB: 3sh5) and bile acid receptor protein (Q96RI1, PDB:3bej) were downloaded from The UniProt Knowledgebase (https://www.uniprot.org/). Molecular Docking simulations of the compounds inside the protein, which from HSPG and bile acid receptor, were carried out by using Molecular Operating Environment (MOE) 2008.10. Initially, structures of compounds were protonated with addition of polar hydrogens, and then converted to three-dimensional (3D) structures, followed by energy minimization with force-field using HyperChem 8.0.7 to get stabilized conformers. The stabled conformer of compounds were introduced to MOE, and then proceeded an “energy minimize” process to offer structurally optimized compounds, and were saved as PDB format files, respectively. After crystal structures of the receptor protein were introduced to MOE, unbound water, other small molecules, and ions were removed, then “protonate 3D” was proceeded, to add proteins to the proteins, subsequently followed by an “energy minimize” process to give structurally optimized protein. A structurally optimized compound was then introduced to optimized protein to proceed docking simulation. Docking score and interaction sites, types, and distances, along with two-dimensional (2D) and 3D interaction diagrams, were obtained with the MOE method. 

## 4. Conclusions

As a part of our series of works for exploring anti-HBV agents, a series of non-nucleoside anti-HBV compounds, by attaching the oximes to dehydrocholic acid (DHCA), have been designed, synthesized, and screened for their bioactivity. Five compounds were found with more effective inhibition of HBeAg than the positive control ETV, two of them, **2b-3** and **2b-1**, showed significant anti-HBV activities against the secretion of HBeAg (IC_50 (**2b-3**)_ = 49.39 ± 12.78, SI**_2b-3_** = 11.03 and IC_50 (**2b-1**)_ = 96.64 ± 28.99 μM, SI**_2b-1_** = 10.35). Results of the molecular docking study showed that there are strong interactions between these compounds and bile acid receptor and HSPG.

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
