# Peer review of "Design, Synthesis and Bioactive Evaluation of Oxime Derivatives of Dehydrocholic Acid as Anti-Hepatitis B Virus Agents"

_molecules, 2020, doi:10.3390/molecules25153359_

Round 1

Reviewer 1 Report

The authors describe the design and bioactive evaluation of oxime derivatives of dehydrocholic acid as potent anti-hepatitis B virus agents. The manuscript will be of interest to readers of Molecule but is best suited for a medicinal chemistry journal. I do recommend publication after addressing the following points. The authors have done a commendable job with the research and presentation. These are some of the points to address which might improve the quality of the manuscript.

The main concern here is the lower activity of the oxime derivatives. Although the synthesized molecules has better activity than entecavir, most have activity in the 100 to 300 micromolar range (withe the exception of 2b-3). So unless there is further exploration, the structure activity relationship is very flat. The authors should make it clear that this is the beginning of a drug discovery campaign with future effort directed towards optimizing the potency. And what do they intend to do with this series.

The Title could read as "Design and bioactive evaluation of oxime derivatives of dehydrocholic acid as anti-hepatitis B virus agents." The oxime derivatives have micromolar activity which is not even in single digit. So use of the word "potent" is not justified.

In the abstract the oximation can be changed to oxime formation.

In the introduction section the sentence "That prompted us and other researchers to invent and find new structural non-nucleoside analogue compounds" compounds need to be rewritten.

The compounds are very well characterized except compound 0-1 which is missing a carboxylic acid proton.

The difference in activity between 2C-3 and 2C-1 is negligible and should be mentioned. From the given data 2C-3 is not better than 2C-1.

DHCA has no activity. In the text change activity from "slight" to none.

Reference 18 20 30 page number needs formatting.

I would request the authors to improve their manuscript presentation and reduce grammatical errors.

Author Response

Dear reviewer

Thank you for your reviewing our manuscript and helpful comments. We have revised our manuscript carefully according to your comments.

Reviewer #1

The authors describe the design and bioactive evaluation of oxime derivatives of dehydrocholic acid as potent anti-hepatitis B virus agents. The manuscript will be of interest to readers of Molecule but is best suited for a medicinal chemistry journal. I do recommend publication after addressing the following points. The authors have done a commendable job with the research and presentation. These are some of the points to address which might improve the quality of the manuscript.

The main concern here is the lower activity of the oxime derivatives. Although the synthesized molecules has better activity than entecavir, most have activity in the 100 to 300 micromolar range (withe the exception of 2b-3). So unless there is further exploration, the structure activity relationship is very flat. The authors should make it clear that this is the beginning of a drug discovery campaign with future effort directed towards optimizing the potency. And what do they intend to do with this series.

Answer:

We have explained in “1. Introduction” part

“ Drugs targeted and concentrated to liver organ will increased their anti-HBV effectivity and decreased side effects. Therefore DHCA, with reactive functional groups, was considered as a liver-target vehicle to deliver drugs to liver in our present work.”

4 Conclusion “As a part of our series works for exploring anti-HBV agents”

The Title could read as "Design and bioactive evaluation of oxime derivatives of dehydrocholic acid as anti-hepatitis B virus agents." The oxime derivatives have micromolar activity which is not even in single digit. So use of the word "potent" is not justified.

Answer: the title have been rewritten as “Design, Synthesis and Bioactive Evaluation of Oxime Derivatives of Dehydrocholic Acid as Anti-Hepatitis B Virus Agents”

In the abstract the oximation can be changed to oxime formation.

Answer: the word “oximation” has been rewritten as “oxime formation”

In the introduction section the sentence "That prompted us and other researchers to invent and find new structural non-nucleoside analogue compounds" compounds need to be rewritten.

Answer: this sentence "That prompted us and other researchers to invent and find new structural non-nucleoside analogue compounds" has been rewritten as “These disadvantages of nucleoside analogs prompted us and other researchers to invent and find new structural non-nucleoside analog compounds”.

The compounds are very well characterized except compound 0-1 which is missing a carboxylic acid proton.

Answer: In solvent DMSO-d, proton in carboxylic acid was deuterated by trace D2O (proton in =NOH is less reactive than that in carboxylic acid, and was not deuterated in the same process).

The difference in activity between 2C-3 and 2C-1 is negligible and should be mentioned. From the given data 2C-3 is not better than 2C-1.

Answer: we have rewritten and explained this result as following:

 “Compound 2c-1, 2c-2, and 2c-3 with similar skeletons but different oxime groups showed different anti-HBV activity and cytotoxicity”was rewritten as “Compound 2c-1, 2c-2, and 2c-3 with similar skeletons but different oxime groups showed slightly different anti-HBV activity and cytotoxicity. Although with lower IC50 values for inhibiting secretion of HBeAg than oxime 2c-1 and benzyl oxime ether 2c-2, Methyl oxime ether 2c-3 was not effective inhibitor because of its cytotoxicity (SI=0.86).”

DHCA has no activity. In the text change activity from "slight" to none.

Answer: it was revised as following

 “3.3. Structure–Activity Relationship (SAR)

DHCA exhibited slightly activity against the secretion of HBeAg and HBsAg”

Has been rewritten as “DHCA exhibited none activity against the secretion of HBeAg and HBsAg”

Reference 18 20 30 page number needs formatting.

Answer: Journals Heliyon (https://www.sciencedirect.com/science/article/pii/S2405844019363984), Bioorg. Med. Chem. Lett (https://www.sciencedirect.com/science/article/pii/S0960894X19305530), and Molecules (https://www.mdpi.com/1420-3049/24/11/2063/htm) don’t provide page number in their articles, but only article numbers. So we are not able to provide page numbers for these papers.

I would request the authors to improve their manuscript presentation and reduce grammatical errors.

Answer:

We have revised and corrected grammatical errors and improved manuscript presentation.

“Docks of the compounds to protein residues from HSPG and bile acid receptor was carried out by using MOE 2008.10” This sentence has been rewritten s “Docks of the compounds to protein residues, which from HSPG and bile acid receptor, were carried out by using MOE 2008.10”

“Ligands date bank of the optimized compounds was built using the builder interface of the MOE program”, this sentence has been rewritten as “Ligands date bank of the optimized compounds were built by using the builder interface of the MOE program”

“Many anti-HBV and non-nucleoside analog compounds have been designed and developed on the basis of their interactions with receptor using molecular docking [18-22].” Have been revised as “ Many anti-HBV bioactive non-nucleoside analog compounds have been designed and developed on the basis of their interactions with receptor using molecular docking [18-22].”

“Table 1 The cytotoxicity and inhibitory effect of target compounds on HBeAg and HBsAg” has ben revised as “Table 1 Cytotoxicity and inhibitory effect of target compounds on HBeAg and HBsAg in vitro

Reviewer 2 Report

The authors are shown the design, synthesis, invitro and insilico evaluation of nove oxime derivatives of dehydrocholic acid, as anti-HBV drug candidates.

The methodology is well designed, following by synthesis and biological evaluation. Finally molecular docking simulations validated the binding mode of the most potent compounds.

However there are some critical issues to be addressed and corrected before acceptance.

First the title should also include the Synthesis.

2.3 Molecular Docking

The methodology of Molecular Docking has to be re-written because the English language needs corrections.

“Docks of the compounds to protein residues from HSPG and bile acid receptor was carried out by using MOE 2008.10”

“Ligands date bank of the optimized compounds was built using the builder interface of the MOE program”

Scheme 1: Labels of compounds are not clear 0-1, 2a-1 etc… need explanations.

Reulsts 3.4. Docking scores are in kcal/mol. Two digits are enough. Those are empirical binding scores and have to be mentioned.

Figure 3. Very bad resolution. Labels are not clear.

Figures 5-8 “Proposal binding….” must be rephrased as “Theoretical Binding mode of compound……”

Table 2 and table 3 should go to supplementary. Also on figures keep only the most potent in superposition with cholic acid or dehydrocholic acid to emphasize the similarity in binding mode.

Author Response

Respond to reviewer

Dear reviewer

Thank you for your reviewing our manuscript and helpful comments. We have revised our manuscript carefully according to your comments.

Reviewer #2

The authors are shown the design, synthesis, invitro and insilico evaluation of nove oxime derivatives of dehydrocholic acid, as anti-HBV drug candidates.

The methodology is well designed, following by synthesis and biological evaluation. Finally molecular docking simulations validated the binding mode of the most potent compounds.

However there are some critical issues to be addressed and corrected before acceptance.

First the title should also include the Synthesis.

Answer:title bas ben rewritten as “Design, Synthesis and Bioactive Evaluation of Oxime Derivatives of Dehydrocholic Acid as Anti-Hepatitis B Virus Agents”

2.3 Molecular Docking

The methodology of Molecular Docking has to be re-written because the English language needs corrections.

“Docks of the compounds to protein residues from HSPG and bile acid receptor was carried out by using MOE 2008.10”

Answer: This sentence has been rewritten s “Docks of the compounds to protein residues, which from HSPG and bile acid receptor, were carried out by using MOE 2008.10”

Other errors have been revised.

“Ligands date bank of the optimized compounds was built using the builder interface of the MOE program”

Answer:

 this sentence has been rewritten as “Ligands date bank of the optimized compounds were built by using the builder interface of the MOE program”

Scheme 1: Labels of compounds are not clear 0-1, 2a-1 etc… need explanations.

Answer:

We have redrawn this scheme to make structure of compounds clear:

Reulsts 3.4. Docking scores are in kcal/mol. Two digits are enough. Those are empirical binding scores and have to be mentioned.

Answer: all docking scores in table 2 and table 3 have been revised in two digits.

Those docking scores were mentioned and in section 3.4.

Figure 3. Very bad resolution. Labels are not clear.

Answer: resolution of Figures 3-8 will be satisfied when they were zoomed in, labels will be clear enough.

Figures 5-8 “Proposal binding….” must be rephrased as “Theoretical Binding mode of compound……”

Answer:

Titles of Figures 3-8 have been rephrased as “Theoretical Binding mode of compound……”

Table 2 and table 3 should go to supplementary. Also on figures keep only the most potent in superposition with cholic acid or dehydrocholic acid to emphasize the similarity in binding mode.

Answer:

Table 2 and table 3 were posited in supplementary. Figure 8 was remove to supplementary too.

Round 2

Reviewer 2 Report

Although most of the suggestions/corrections are re-written i found some critical errors in English, such as:

2.3. Molecular Docking

The phrase "Ligands date bank of the optimized compounds were built by using the builder interface of the MOE program " doesn;t make sense because Lgadns date bank is wrong. That was my first indication when i suggested to rephrase the sentence. 

Additionally the third line start wrongly. "Docs of the compounds" should be "Molecular Docking simulations of the compounds inside the protein"

Please make the corrections in order for the manuscript to be well presetned and accepted. 

Author Response

Responds to reviewer

Dear reviewer

Thank you very much to single out mistakes and errors in our manuscript. We have revised and rewritten related contents to make it clear.

Thank you.

With best regards

Correspondent author

Dr. Wanxing Wei

2.3. Molecular Docking

1# The phrase "Ligands date bank of the optimized compounds were built by using the builder interface of the MOE program " doesn;t make sense because Lgadns date bank is wrong. That was my first indication when i suggested to rephrase the sentence. 

2# Additionally the third line start wrongly. "Docs of the compounds" should be "Molecular Docking simulations of the compounds inside the protein"

 This phrase has been rewrite according to reviewer’s comment

Please make the corrections in order for the manuscript to be well presetned and accepted. 

Revision and correction:

1#

Before revised:

Initially the structures of the compounds were protonated with addition of polar hydrogens and then converted to 3D structure followed by energy minimization with force-field using the HyperChem 8.0.7 to get stabilized conformer of the protein. Ligands date bank of the optimized compounds were built by using the builder interface of the MOE program. The crystal structures were optimized in MOE, including addition of polar hydrogens for protonation, removal of unbound waters, and 3D structure of pronation, and then docking with the optimized ligands. Geometry of the resulting complexes was studied using the MOE’s Pose Viewer utility.

After revised:

Initially structures of compounds were protonated with addition of polar hydrogens and then converted to 3D structure followed by energy minimization with force-field using the HyperChem 8.0.7 to get stabilized conformer. These stabled conformer of compounds were introduced to MOE, and then proceeded “energy minimize” process to offer structurally optimized compounds and saved as pdb format file respectively. After crystal structures of receptor’s protein were introduced to MOE, unbound water, other small molecules and ions were removed, then “protonate 3D” was proceeded to add proteins to the proteins, subsequently followed “energy minimize” process to give structurally optimized protein. Structurally optimized compound was then introduced to optimized protein to proceed docking simulation. Docking score and interaction sites, types and distances along with 2D and 3D interaction diagrams were obtained with the MOE method.   

2#

This phrase has been rewrite according to reviewer’s comment.